# GROUP VERIFICATION-BASED POLICY OPTIMIZATION FOR INTERACTIVE CODING AGENTS

**Silong Dai**[*1,2#]**, Changzhi Sun** [*2,†]**, Haolun Wu**[1]**, Huanran Zheng**[1]**, Tao Ji**[3]**, Junchi Yan**[4]**,**
**Yuanbin Wu**[1,†]**, Dell Zhang**[2]**, Xiaoling Wang**[1,†]**, Xuelong Li**[2,†]

[1]Department of Computer Science and Technology, East China Normal University
[2]Institute of Artificial Intelligence (TeleAI), China Telecom
[3]College of Foreign Languages and Literatures, Fudan University
[4]School of Artificial Intelligence, Shanghai Jiao Tong University
{sldai@stu, ybwu@cs, xlwang@cs}.ecnu.edu.cn
{dell.z, xuelong_li}@ieee.org

## ABSTRACT

Recent advancements in reinforcement learning from verifiable rewards (RLVR), particularly through Group Relative Policy Optimization (GRPO), have significantly improved the capabilities of large language models (LLMs) for interactive coding agents. However, these methods overlook process-verifiable environment feedback (e.g., code execution failures), leading to inaccurate advantage estimation at each reasoning step and insufficient learning. To address this issue, we propose Group Verification-based Policy Optimization (GVPO), a novel RL algorithm that introduces an advantage shaping framework integrating both outcome-verifiable and process-verifiable signals. While outcome-verifiable rewards ensure alignment with long-term task objectives, process-verifiable feedback derived from intermediate execution traces (e.g., syntax errors, runtime exceptions) serves as corrective shaping terms at the step level. By jointly leveraging these two forms of verifiability, GVPO achieves more accurate credit assignment, balancing short-term process guidance with long-term outcome alignment. This unified formulation yields more stable optimization, faster convergence, and stronger generalization in complex interactive environments. A 32B-parameter agent trained with GVPO in the AppWorld environment outperforms OpenAI's o1 agent by 12.7% on the more challenging Test-C split and surpasses the strongest 32B RL-trained state-of-the-art baseline by 3.7%.

## 1 INTRODUCTION

Large language models (LLMs) have recently demonstrated remarkable progress in understanding, reasoning, embodied interaction, and code generation, positioning them as promising candidates for interactive coding agents (Chen et al., 2025b; Li et al., 2024; Song et al., 2025; Chen et al., 2025a), and aligning with the broader vision of AI Flow that emphasizes scalable and collaborative intelligence (An et al., 2026). A key challenge, however, lies in training these agents to operate reliably in complex environments where they must engage in multi-turn interactions, plan dynamically, and generate executable code to achieve user-specified goals. Reinforcement learning from verifiable rewards (RLVR) has emerged as a powerful paradigm to address this challenge (Guo et al., 2025), as it enables scalable supervision without costly human annotations by leveraging deterministic signals. Among RLVR approaches, Group Relative Policy Optimization (GRPO) (Shao et al., 2024) has proven particularly effective, significantly advancing the performance of LLM-based agents (Li et al., 2025; Yu et al., 2025).

Despite these successes, current RLVR methods still exhibit critical limitations. Most notably, they rely almost exclusively on **outcome-verifiable rewards**, such as exact answer matching. While such

---

*Equal contribution; # Work done while this author was an intern at TeleAI; † Corresponding authors.

| Method | Clip | Aggr. | Advantage Function | | | | | |
|--------|------|-------|--------|-------|------|---------|------|-------|
| | | | Calcu. | Traj. | Step | Unbias. | Out. | Proc. |
| RLOO (Ahmadian et al., 2024) | *Sym.* | *smtm* | GR | ✓ | ✗ | ✓ | ✓ | ✗ |
| GRPO (Shao et al., 2024) | *Sym.* | *smtm* | GR | ✓ | ✗ | ✓ | ✓ | ✗ |
| Dr.GRPO (Liu et al., 2025) | *Sym.* | *smts* | GR | ✓ | ✗ | ✓ | ✓ | ✗ |
| DAPO (Yu et al., 2025) | *Asy.* | *tm* | GR | ✓ | ✗ | ✓ | ✓ | ✗ |
| LOOP (Chen et al., 2025b) | *Sym.* | *smtm* | GR | ✓ | ✗ | ✓ | ✓ | ✗ |
| GVPO (Ours) | *Asy.* | *smtm* | Shaping | ✓ | ✓ | ✗ | ✓ | ✓ |

Table 1: Comparison of RLVR methods in LLMs. **Clip**: *Sym.* = symmetric clipping (single $\epsilon$ for both sides); *Asy.* = asymmetric clipping (separate high/low bounds, with explicit "clip-high" control). **Aggr.**: loss aggregation scheme, where *smtm* = sequence-mean-token-mean, *smts* = sequence-mean-token-sum, and *tm* = token-mean. **Calcu.** (Calculation): GR = group-relative advantage; Shaping = outcome+process advantage shaping. Advantage Function: **Traj.** = trajectory-level; **Step** = step-level. **Unbias.** = whether the estimator preserves the unbiased (zero-mean) property. **Out.** = whether outcome-verifiable rewards are incorporated; **Proc.** = whether process-verifiable signals are incorporated. GVPO is the only method that integrates both outcome- and process-level rewards through advantage shaping, achieves step-level credit assignment, and employs asymmetric clipping.

rewards faithfully capture task-level correctness, they are inherently sparse and delayed, offering little guidance during the intermediate steps of reasoning. As a result, *credit assignment* becomes inaccurate: early-stage errors may still receive positive reinforcement if the final outcome succeeds, while partially correct reasoning may be discarded if the trajectory ultimately fails. This issue leads to unstable optimization, slow convergence, and underutilization of valuable environment feedback.

One underexplored direction is the integration of **process-verifiable signals**, intermediate feedback derived from execution traces such as syntax errors, runtime exceptions, or partial unit test results. Unlike outcome-based signals, process feedback is **dense, fine-grained, and deterministic**, providing rich supervision at the token or step level. However, existing RLVR methods (Yu et al., 2025), including GRPO and its variants (Liu et al., 2025), do not incorporate such signals into their learning framework, thereby missing opportunities for more precise credit assignment and error correction.

In this paper, we introduce Group Verification-based Policy Optimization (GVPO), a novel reinforcement learning algorithm that addresses this gap through an **advantage shaping framework**. GVPO extends group-based policy optimization by integrating both outcome-verifiable and process-verifiable signals into the advantage function. Specifically, outcome-verifiable rewards ensure that learning remains aligned with long-term task objectives, while process-verifiable signals act as corrective shaping terms that adjust step-level credit assignment in real time. This design mitigates the risk of reinforcing error-prone reasoning patterns and amplifies partial successes, effectively balancing short-term guidance with long-term alignment. Tab. 1 presents a comparison of RLVR methods in LLMs.

We validate GVPO in AppWorld, a challenging benchmark environment where agents must solve long-horizon, multi-turn tasks spanning multiple applications and APIs. Our experiments show that a **32B-parameter agent trained with GVPO** outperforms OpenAI's o1 agent by 12.6% on the difficult Test-C split and surpasses the strongest 32B RL-trained state-of-the-art baseline by 3.6%. These results establish GVPO as a new milestone for RLVR-based training of interactive coding agents. In summary, this work makes the following contributions:

- We identify the limitations of current RLVR approaches that rely solely on outcome-verifiable rewards and highlight the importance of integrating process-verifiable signals for precise credit assignment.
- We propose GVPO, the RL algorithm that unifies outcome-verifiable and process-verifiable signals through an advantage shaping framework.
- We demonstrate through extensive experiments in AppWorld that GVPO yields substantial improvements in stability, convergence, and overall performance, outperforming both open-source and closed-source baselines.

## 2 PRELIMINARY

### 2.1 GROUP RELATIVE POLICY OPTIMIZATION (GRPO)

GRPO estimates the advantage in a group-relative manner. We denote a user request as a natural language instruction $q$. The behavior policy $\pi_{\theta_{\text{old}}}$ samples a group of $G$ individual responses $\{\tau_i\}_{i=1}^G$. The advantage of the $i$-th response is then computed by normalizing its group-level reward within the sampled set:

$$\hat{A}_{i,t} = \frac{R_i - \text{mean}(\{R_i\}_{i=1}^G)}{\text{std}(\{R_i\}_{i=1}^G)}$$

Note that for a response $\tau_i$, the computation of $\hat{A}_{i,t}$ is independent of $t$; that is, all timesteps $t$ share the same advantage value. GRPO then optimizes a clipped objective:

$$\mathcal{J}_{\text{GRPO}}(\theta) = \frac{1}{G} \sum_{i=1}^G \frac{1}{|\tau_i|} \sum_{t=1}^{|\tau_i|} \left\{ \min \left[ a_{i,t}(\theta)\hat{A}_{i,t}, \, \text{clip}\left(a_{i,t}(\theta), 1-\epsilon, \, 1+\epsilon\right) \hat{A}_{i,t} \right] \right\},$$

$$a_{i,t}(\theta) = \frac{\pi_\theta(\tau_{i,t} \mid q, \tau_{i,<t})}{\pi_{\theta_{\text{old}}}(\tau_{i,t} \mid q, \tau_{i,<t})},$$

where $\epsilon$ is a hyperparameter. In this work, we drop the KL term; beyond reducing the computational and memory cost of maintaining $\pi_{\text{ref}}$, this choice is also supported by recent evidence that it can enhance R1-Zero–style training (Liu et al., 2025).

### 2.2 INTERACTIVE CODING AGENTS

Interactive Coding Agents (ICA) represent a new paradigm of intelligent agents that accomplish tasks through iterative interaction with external APIs by executing code snippets (e.g., Python). Similar to ReAct (Yao et al., 2023), ICA decomposes the process into two components: *reasoning* and *action*. However, unlike ReAct where actions are natural language commands, ICA employs *executable code* as actions, making it more suitable for tool-rich environments (Fig. 4).

Formally, let $\pi_\theta$ denote a language model, let $I$ be a code interpreter that executes code written by $\pi_\theta$ to realize tool calls, and let $q$ be an input query. Let $\tau$ denote the $i$-th sampled trajectory; for notational simplicity we suppress the subscript $i$. We construct the partial reasoning trajectory at step $k$ as [1]:

$$\tau[k] = r_1, c_1, o_1, \cdots, r_k, c_k, o_k,$$

where $r_j$ denotes natural language reasoning, $c_j$ denotes generated code, and $o_j$ is the execution result of $c_j$. The iterative generation process for trajectory $\tau$ follows:

$$(r_k, c_k) = \pi_\theta(\cdot \mid q \oplus \tau[k-1]), \quad o_k = I(c_k), \quad \tau[k] = \tau[k-1] \oplus r_k \oplus c_k \oplus o_k.$$

Here, $\oplus$ indicates sequence concatenation. This cycle continues until the model produces a final answer or the maximum number of reasoning steps $K_{\max}$ is reached, with each step informed by previous code-execution results.

Next, we clarify some notation to facilitate the subsequent exposition.

**Definition 1** (Index sets over a trajectory). *Given a trajectory $\tau_i$, we define $\mathcal{I}_i = \{1, \ldots, |\tau_i|\}$. For each $j \in \mathcal{I}_i$, let $\tau_{i,j}$ denote the $j$-th token in trajectory $\tau_i$. We partition $\mathcal{I}_i$ into two disjoint subsets:*

$$\mathcal{I}_i^G \cup \mathcal{I}_i^O = \mathcal{I}_i, \quad \mathcal{I}_i^G \cap \mathcal{I}_i^O = \emptyset,$$

*where $\mathcal{I}_i^G$ collects the indices of generation tokens (including both reasoning and code tokens), and $\mathcal{I}_i^O$ collects the indices of observation tokens (execution feedback returned by the interpreter). In addition, we introduce two index sets:*

$$\mathcal{I}_i^{\text{succ}} \cup \mathcal{I}_i^{\text{fail}} = \mathcal{I}_i^G, \quad \mathcal{I}_i^{\text{succ}} \cap \mathcal{I}_i^{\text{fail}} = \emptyset.$$

*The set $\mathcal{I}_i^{\text{succ}}$ collects the indices of reasoning and code tokens whose corresponding step execution succeeded (no error message), while $\mathcal{I}_i^{\text{fail}}$ collects those whose execution failed (with error messages). Observation indices $\mathcal{I}_i^O$ are not included in this split. Thus, every reasoning/code token in the trajectory belongs either to $\mathcal{I}_i^{\text{succ}}$ or to $\mathcal{I}_i^{\text{fail}}$, but not both.*

---

[1]We distinguish between *timestep* $t$, which denotes the $t$-th generated token (token-level granularity), and *reasoning step* $k$, which denotes the $k$-th reasoning cycle in ICA, consisting of reasoning, code generation, and execution ($r_k, c_k, o_k$).

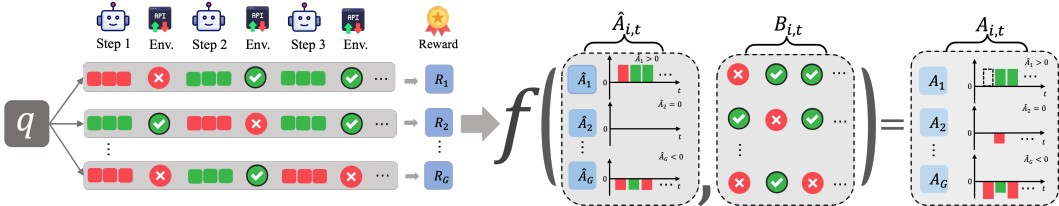

Figure 1: Overview of the proposed GVPO. For each question $q$, multiple trajectories are sampled to interact with the environment and yield outcome-based rewards $R_i$. Outcome-verifiable advantages $\hat{A}_{i,t}$ are derived from these rewards, while process-verifiable feedback $B_{i,t}$ captures step-level successes and failures. A shaping function $f(\cdot)$ integrates both signals to produce the final advantages $A_{i,t}$, enabling outcome alignment with step-level correction.

## 3  APPROACH

In this section, we first present the proposed algorithm (Sec. 3.1), followed by a description of the outcome-based reward functions (Sec. 3.2) and the process-verifiable functions (Sec. 3.3) within the AppWorld environment. An overview of GVPO is illustrated in Fig. 1.

### 3.1  GROUP VERIFICATION-BASED POLICY OPTIMIZATION (GVPO)

For each question $q$, GVPO samples a group of outputs $\{\tau_i\}_{i=1}^{G}$, and optimizes the policy via the following objective:

$$\mathcal{J}_{\text{GVPO}}(\theta) = \frac{1}{G} \sum_{i=1}^{G} \frac{1}{|\tau_i|} \sum_{t=1}^{|\tau_i|} \left\{ \min\left[ a_{i,t}(\theta) A_{i,t}, \text{clip}\left( a_{i,t}(\theta), 1 - \epsilon_{\text{low}}, 1 + \epsilon_{\text{high}} \right) A_{i,t} \right] \right\}, \quad (1)$$

$$a_{i,t}(\theta) = \frac{\pi_\theta(\tau_{i,t} \mid q, \tau_{i,<t})}{\pi_{\theta_{\text{old}}}(\tau_{i,t} \mid q, \tau_{i,<t})}.$$

**Loss Aggregation.**  The objective in Eq. 1 aggregates learning signals hierarchically across both the group and sequence dimensions. GVPO adopts the sequence-mean-token-mean (*smtm*) scheme: within each trajectory, token-level contributions are first averaged to obtain a token mean, and these are then averaged across sequences to yield the final objective.

For comparison, GRPO also employs *smtm*, whereas Dr.GRPO (Liu et al., 2025) uses the sequence-mean-token-sum (*smts*) scheme, in which token contributions are summed within each sequence before averaging across sequences, thereby amplifying the influence of longer trajectories. An alternative variant, *smtm*, averages at the sequence level first and then distributes uniformly to tokens, yielding a more length-invariant signal. By contrast, DAPO (Yu et al., 2025) applies a simpler token-mean (*tm*) strategy, directly averaging token-level signals without additional sequence-level aggregation.

**Advantage Shaping.**  We introduce the notion of an advantage shaping function, which integrates both outcome-verifiable and process-verifiable signals into the advantage estimation:

$$A_{i,t} = f(\hat{A}_{i,t}, B_{i,t}), \quad \hat{A}_{i,t} = R_i - \text{mean}(\{R_i\}_{i=1}^{G}).$$

where $\hat{A}_{i,t}$ is the outcome-verifiable advantage, defined relative to group-based rewards without std normalization terms (Liu et al., 2025), and $B_{i,t}$ denotes process-level feedback derived from execution signals (e.g., compilation status, runtime exceptions, or partial unit-test outcomes). The shaping function $f(\cdot)$ provides a general mechanism to modulate $\hat{A}_{i,t}$ using deterministic process feedback, thereby calibrating the policy's credit assignment. It is important to note that after shaping, $A$ no longer preserves the unbiased property (i.e., $\mathbb{E}[A] = 0$ no longer holds).

**Intuition.**  Outcome-verifiable rewards ensure alignment with final task objectives, while process-verifiable signals serve as corrective shaping terms that guide learning at a finer granularity. When

execution feedback indicates early-stage failures, $f(\cdot)$ introduces negative corrections to reduce the likelihood of reinforcing error-prone patterns. Conversely, partial successes yield positive corrections, amplifying behaviors that show promising progress before final outcomes are observed.

By shaping the advantage with both outcome-level and process-level information, the policy benefits from more accurate credit assignment across trajectories. This unified formulation ensures that learning is not only guided by final correctness but also by intermediate execution quality, leading to more stable optimization, faster convergence, and stronger performance in complex problem-solving environments.

## 3.2 OUTCOME-VERIFIABLE REWARD FUNCTIONS

In AppWorld, each task is associated with a set of unit tests that check whether the agent's generated code correctly produces the desired state changes without introducing unintended side effects. We leverage these unit tests to construct an outcome-based reward signal (Chen et al., 2025b).

Formally, for a trajectory $\tau_i$ produced in response to query $q$, let $\{u_j\}_{j=1}^M$ denote the $M$ unit tests associated with the task. Each unit test $u_j$ returns a binary result $\text{pass}(u_j, \tau_i) \in \{0, 1\}$, indicating whether the output passes the test. The outcome-based reward is defined as the fraction of passed tests:

$$R_i = \frac{1}{M} \sum_{j=1}^M \text{pass}(u_j, \tau_i), \tag{2}$$

where $R_i \in [0, 1]$. This scalar reward provides a direct measure of the final correctness of the generated output. It is sparse in nature—nonzero signals are only available once execution has completed—but it captures the ultimate task objective faithfully.

## 3.3 PROCESS-VERIFIABLE FUNCTIONS

While outcome-based rewards provide only sparse signals at the end of execution, process-verifiable functions deliver dense step-level feedback by differentiating between successful and failed reasoning/code tokens within a trajectory $\tau_i$. Formally, let $\mathcal{I}_i^{\text{succ}}$ and $\mathcal{I}_i^{\text{fail}}$ denote the index sets corresponding to successful and failed reasoning/code tokens, respectively. Based on these index sets, we define a process-verifiable bonus $B_{i,t}$ that adaptively adjusts the step-level advantage $A_{i,t}$:

$$B_{i,t} = \begin{cases} 0 & t \in \mathcal{I}_i^{\text{succ}}, \\ -b & t \in \mathcal{I}_i^{\text{fail}} \wedge \hat{A}_{i,t} = 0, \\ b \cdot \hat{A}_{i,t} & t \in \mathcal{I}_i^{\text{fail}} \wedge \hat{A}_{i,t} < 0, \\ -\hat{A}_{i,t} & \text{otherwise}. \end{cases} \iff A_{i,t} = \begin{cases} \hat{A}_{i,t} & t \in \mathcal{I}_i^{\text{succ}}, \\ \hat{A}_{i,t} - b & t \in \mathcal{I}_i^{\text{fail}} \wedge \hat{A}_{i,t} = 0, \\ (1+b) \cdot \hat{A}_{i,t} & t \in \mathcal{I}_i^{\text{fail}} \wedge \hat{A}_{i,t} < 0, \\ 0 & \text{otherwise}. \end{cases}$$

Here, $b > 0$ is a fixed penalty coefficient (set to $b = 0.2$ in our experiments). The adjustment mechanism can be interpreted as follows:

- **Successful tokens** (those associated with correct reasoning or code execution) retain their outcome-based advantage $\hat{A}_{i,t}$ unchanged.
- **Failed tokens** are penalized in a manner sensitive to the trajectory's outcome advantage:
  - When $\hat{A}_{i,t} = 0$, a fixed penalty $-b$ is imposed.
  - When $\hat{A}_{i,t} < 0$, the negative outcome advantage is amplified proportionally.
  - In all other cases, the advantage is set to zero.
- **Observation tokens** represent environment feedback and are excluded from updates, hence set to zero. (Li et al., 2025).

When the advantage is negative, we apply a multiplicative adjustment to penalize failed trajectories, while also exploring an additive variant in ablations. This yields more robust credit assignment under noisy or misleading signals.

**Scalability.** An important advantage of the process-verifiable paradigm is that it is inherently rule-based and does not require training an additional reward model. This property makes it highly scalable: whenever the environment provides deterministic feedback signals, such as compilation status, runtime errors, constraint checks, or state-transition validations, process-verifiable rewards can be applied directly. While we instantiate this idea in the coding domain (AppWorld), the paradigm is not limited to program synthesis. Any environment capable of emitting reliable process-level feedback can naturally support this form of supervision, enabling efficient scaling across diverse tasks without costly human annotation or reward-model training.

## 4 EXPERIMENTS

**Dataset.** We evaluate on AppWorld (Trivedi et al., 2024), a benchmark for interactive coding agents that requires multi-turn planning and executable code generation in a stateful Python environment. It integrates nine simulated consumer apps (e.g., email, payments, shopping, file system), exposing $457$ APIs across realistic digital activities. The benchmark defines $750$ tasks from $250$ scenarios, split into train (35/105), dev (20/60), test-normal (56/168), and test-challenge (139/417), with the latter involving unseen apps and more complex planning.

**Evaluation.** We report results using two key metrics: Task Goal Completion (TGC), the percentage of tasks in which the agent passes all evaluation tests, and Scenario Goal Completion (SGC), the percentage of scenarios in which the agent succeeds on every associated task.

**Implementation.** We adopt Qwen2.5-32B-Instruct as the base model and train it with the veRL (Sheng et al., 2024) framework, using vLLM (Kwon et al., 2023) for efficient batched inference. Training is restricted to difficulty-1/2 tasks in AppWorld (72 samples, 24 scenarios) with 8 rollouts per sample, a maximum of 40 interaction turns for training and 50 for evaluation, temperature 1.0 during training (exploration) and 0 at evaluation (deterministic execution). More details can be found in Appendix A.1.

**Baselines.** We compare GVPO against both zero-shot LLMs (GPT-4o, GPT-4 Trb, Llama-3 70B, OpenAI o1, and Qwen2.5-32B-Instruct) and RL-trained models (RLOO, GRPO, GSPO, Dr.GRPO, and LOOP), all optimized with unit-test–based rewards $R_i \in [0, 1]$. We select the checkpoint achieving the highest TGC score on the development set. Detailed descriptions of the baselines are provided in Appendix A.2.

### 4.1 RESULT

**Main Results.** Tab. 2 compares zero-shot prompting models and RL fine-tuned methods on the AppWorld benchmark. Among the zero-shot models, OpenAI o1 achieves the highest performance (61.9 TGC / 41.1 SGC on Test-N), but still struggles on the more challenging Test-C split. RL fine-tuning consistently improves performance over prompting. In particular, our method, GVPO, sets a new state-of-the-art across both splits, achieving 72.6 TGC / 55.4 SGC on Test-N and 49.4 TGC / 28.8 SGC on Test-C. Notably, GVPO surpasses the previously strongest 32B RL baseline, LOOP, by 3.7 points TGC and 2.2 points SGC on Test-C, demonstrating the effectiveness of incorporating process-verifiable signals for credit assignment. While LOOP performs competitively on Test-N, it falls short on Test-C, highlighting GVPO's superior generalization to unseen apps and longer multi-step planning tasks. More analyses of the validation set results can be found in Appendix A.3.

**Entropy Trajectories.** Fig. 2a compares the entropy trajectories of GRPO, DAPO, GSPO, and GVPO during training. GSPO shows the fastest entropy decay, collapsing to a near-deterministic policy, which indicates premature convergence and insufficient exploration.

GRPO and DAPO both maintain moderate entropy levels, but their trajectories still decline steadily, suggesting exploration diminishes over time and leading to potential suboptimal local minima. In contrast, GVPO consistently preserves higher entropy and avoids collapse, reflecting its ability to sustain exploration throughout training. This stability stems from its integration of process-verifiable shaping and asymmetric clipping, which penalize incorrect trajectories without discouraging diver-

| | Para. | Std. | Aggr. | Out. | Proc. | Test-N | | Test-C | |
|---|---|---|---|---|---|---|---|---|---|
| | | | | | | TGC | SGC | TGC | SGC |
| *Prompting with LLM* | | | | | | | | | |
| GPT-4o | - | - | - | - | - | 48.8 | 32.1 | 30.2 | 13.0 |
| GPT-4 Trb | - | - | - | - | - | 26.8 | 12.5 | 17.5 | 5.8 |
| OpenAI o1 | - | - | - | - | - | 61.9 | 41.1 | 36.7 | 19.4 |
| LlaMA3 | 70B | - | - | - | - | 24.4 | 17.9 | 7.0 | 4.3 |
| Qwen2.5 | 32B | - | - | - | - | 34.5 | 16.1 | 18.9 | 7.9 |
| *Fine-tuning with RL* | | | | | | | | | |
| GRPO$_{\text{w/ kl}}$ | 32B | ✓ | *smtm* | ✓ | ✗ | 62.5 | 42.9 | 38.6 | 25.2 |
| GRPO | 32B | ✓ | *smtm* | ✓ | ✗ | 61.3 | 39.3 | 38.5 | 21.6 |
| GSPO | 32B | ✓ | *smtm* | ✓ | ✗ | 56.5 | 33.9 | 37.6 | 17.3 |
| DAPO | 32B | ✓ | *tm* | ✓ | ✗ | 57.1 | 35.7 | 38.8 | 18.0 |
| Dr.GRPO | 32B | ✗ | *smts* | ✓ | ✗ | 63.7 | 44.6 | 40.5 | 18.7 |
| RLOO$_{\text{w/ kl}}$ | 32B | ✗ | *smtm* | ✓ | ✗ | 60.7 | 39.3 | 40.1 | 20.9 |
| LOOP | 32B | ✗ | *smtm* | ✓ | ✗ | 71.3 | 53.6 | 45.7 | 26.6 |
| GVPO | 32B | ✗ | *smtm* | ✓ | ✓ | **72.6** | **55.4** | **49.4** | **28.8** |

Table 2: Test performance is reported on both the normal (Test-N) and challenge (Test-C) splits of AppWorld, using **TGC** (Task Goal Completion) and **SGC** (Scenario Goal Completion). **Para.**: model parameter scale. **Std.**: whether std normalization is applied in advantage computation. **Aggr.**: loss aggregation scheme, where *smtm* = sequence-mean-token-mean, *smts* = sequence-mean-token-sum, and *tm* = token-mean. **Out.**: use of outcome-verifiable signals. **Proc.**: use of process-verifiable signals. GRPO$_{\text{w/ kl}}$ denotes GRPO trained with KL regularization. In DAPO, we omit dynamic sampling and overlong reward shaping. The best results are shown in bold, and the second-best results are underlined.

sity. As a result, GVPO achieves a better balance between exploration and exploitation than GRPO, DAPO, and GSPO, contributing to its superior robustness and final performance.

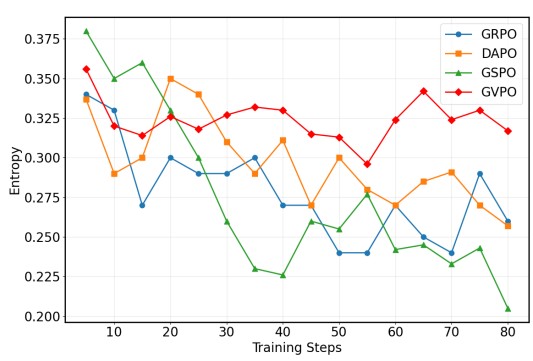

(a) Entropy trajectories of GRPO, DAPO, GSPO, and GVPO during training.

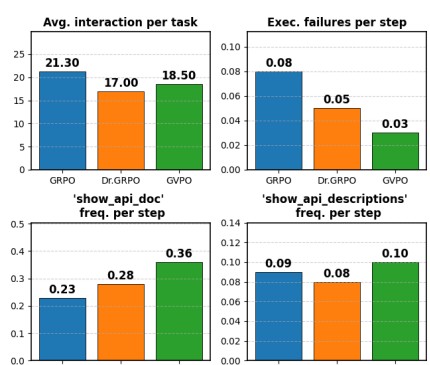

(b) Aggregate changes in agent behavior between GRPO, Dr.GRPO, and GVPO, averaged over three i.i.d. rollouts per dev task.

Figure 2: Training dynamics and behavioral comparisons across RL methods.

## 4.2 ANALYSIS

**Shaping Coefficient** $b$. We further investigate the effect of the shaping coefficient b, which controls the relative strength of step-level penalty in GVPO. To analyze its sensitivity, we conduct a hyperparameter sweep over $b \in \{0.1, 0.2, 0.4\}$ on both Test-N and Test-C benchmarks, and report the results in Tab. 3.

| | Test-N | | Test-C | |
|---|---|---|---|---|
| $b$ | **TGC** | **SGC** | **TGC** | **SGC** |
| 0.1 | 66.7 | 39.3 | 53.1 | 30.9 |
| 0.2 | 72.6 | 55.4 | 49.4 | 28.8 |
| 0.4 | 67.9 | 44.6 | 48.0 | 28.1 |

Table 3: Sensitivity analysis of the shaping coefficient $b$ in GVPO on Test-N and Test-C. Performance comparison under different values of $b$, showing that $b = 0.2$ achieves the best overall and most stable results across both benchmarks.

Overall, we observe that $b = 0.2$ achieves the best and most consistent performance across all settings, and is therefore adopted in our main experiments. When $b = 0.1$, the step-level correction becomes too weak, leading to insufficient credit assignment and degraded scenario-level performance. In contrast, a larger value $b = 0.4$ tends to over-penalize failed intermediate steps, which introduces mild training instability and does not yield further improvement in final task success.

Across the evaluated range, GVPO exhibits relatively stable performance, indicating that the method is moderately sensitive to $b$ and does not rely on fine-grained tuning of this hyperparameter.

| | Setting | Dev | | Test-N | | Test-C | |
|---|---|---|---|---|---|---|---|
| | | **TGC** | **SGC** | **TGC** | **SGC** | **TGC** | **SGC** |
| GVPO | - | **84.2** | **73.7** | **72.6** | **55.4** | **49.4** | **28.8** |
| ***Aggr.*** | *tm* | 73.7 | 52.6 | 58.3 | 35.7 | 34.6 | 15.1 |
| | *smts* | 75.4 | 63.6 | 64.9 | 46.4 | 42.8 | 23.7 |
| ***Clip*** | *Sym.* | 71.9 | 52.6 | 60.1 | 39.3 | 40.5 | 21.6 |
| ***Std.*** | ✓ | 77.2 | 68.4 | 70.8 | 48.2 | 42.8 | 23.7 |
| ***Other*** | *mbs ≠ gs* | 77.2 | 62.5 | 66.7 | 42.9 | 43.1 | 25.9 |
| | *additive shaping only* | 73.7 | 57.9 | 64.9 | 46.4 | 39.8 | 23.0 |

Table 4: Ablation study of GVPO. Performance under different design choices is reported on the AppWorld Dev, Test-N, and Test-C splits. Variants include alternative loss aggregation schemes (*tm*=token-mean, *smts*=sequence-mean-token-sum), symmetric vs. asymmetric clipping, std normalization (*Std.*), and additional settings such as mismatched micro-batch/group sizes (*mbs ≠ gs*), and *additive shaping only* (i.e., $A_{i,t} = \hat{A}_{i,t} - b$, $t \in \mathcal{I}_i^{\text{fail}} \wedge \hat{A}_{i,t} < 0$).

**Ablation Result.** Tab. 4 presents the ablation results of GVPO across the AppWorld dev, Test-N, and Test-C splits. The full GVPO consistently achieves the strongest performance, confirming the effectiveness of its combined design. Alternative aggregation schemes (token-mean, sequence-mean-token-sum) lead to noticeable drops in Test-C, suggesting that GVPO's sequence-mean-token-mean formulation provides a better trade-off between stability and credit assignment. Symmetric clipping underperforms, validating the benefit of the asymmetric "clip-higher" strategy for maintaining exploration. Similarly, applying std normalization (*Std.*) degrades performance, suggesting that removing variance scaling avoids optimization bias and better preserves the reward signal.

For mismatched micro-batch/group sizes (*mbs=8, gs=6*), we observe a larger performance drop here compared to prior math-reasoning tasks, likely because AppWorld tasks involve diverse API calls and state transitions where imbalance in batch-wise normalization amplifies variance in gradient estimates, making training less stable.

In GVPO, for failed tokens with negative outcome advantages we apply multiplicative shaping, i.e., $A_{i,t} = (1+b) \cdot \hat{A}_{i,t}$ for $t \in \mathcal{I}_i^{\text{fail}} \wedge \hat{A}_{i,t} < 0$. In the **additive-only** variant, this scaling is replaced by a constant penalty, i.e., $A_{i,t} = \hat{A}_{i,t} - b$ for the same index set. This substitution weakens the balance between outcome- and process-level signals and leads to poorer generalization, underscoring that the choice of advantage shaping is critical; we leave a deeper investigation to future work.

**Task:** Label all email threads in my Gmail inbox from notifications@<app>.com with the label of the respective app. Ignore spam and archived ones.

| GRPO | Dr.GRPO | GVPO |
|---|---|---|
| Read apps & api descriptions | Check account & password | Read apps & api descriptions |
| ✗ Gmail login with fake password | ✗ Gmail login with wrong api | Check account & password |
| Check account & password | Read apps & api descriptions | Read apps & api descriptions |
| ✗ Gmail login with wrong para. | Read api docs | ✓ Gmail login |
| Read api docs | ✓ Gmail login | Read api docs |
| ✓ Gmail login | Read api docs | ✓ Check email |
| Read api docs | ✓ Check email | Read api docs |
| ⋮ | ⋮ | ✓ Label email |

Figure 3: Case study on the Gmail labeling task. Compared with GRPO and Dr.GRPO, the GVPO-trained agent exhibits a more cautious strategy by extensively consulting API descriptions and documentation before executing concrete actions.

**Agent Behaviors.** Fig. 2b summarizes the behavioral characteristics of agents trained with GRPO, Dr.GRPO, and GVPO across four measures: (i) average number of interactions per task, (ii) execution failure probability per step, (iii) frequency of `show_api_docs` calls, and (iv) frequency of `show_api_descriptions` calls. The results reveal clear behavioral differences among the methods. GVPO achieves the lowest failure probability, while simultaneously exhibiting the highest frequency of documentation queries. In fact, nearly half of the steps taken by GVPO agents involve consulting either `show_api_docs` or `show_api_descriptions`, suggesting that step-level penalties discourage risky or poorly informed actions and instead incentivize information gathering before execution. This behavior reflects a clear shift toward a "query-before-act" strategy: as invalid or high-risk actions are explicitly penalized at the step level, documentation-query actions gain relative utility, reducing the need for corrective loops caused by earlier mistakes. Importantly, this more cautious decision-making does not incur excessive interaction overhead. GVPO requires slightly more steps than Dr.GRPO but fewer than GRPO, which often accumulates longer trajectories due to repeated error recovery.

From a credit assignment perspective, fine-grained process feedback allows GVPO to identify and penalize specific failure modes more precisely than scenario-level rewards alone, enabling more targeted policy updates. Overall, these results indicate that GVPO suppresses unsafe behavior, promotes structured information seeking, and shifts the agent toward more deliberate, verify-then-act reasoning, while preserving sufficient interaction efficiency and diversity.

**Case Study.** Building on these quantitative findings, we next provide a qualitative case study in Fig. 3. In contrast to GRPO and Dr.GRPO, which often make invalid attempts such as logging into Gmail with incorrect parameters or invoking the wrong API, the GVPO adopts a markedly more cautious strategy. Nearly half of its tool invocations are devoted to querying documentation before committing to concrete actions (e.g., Gmail login, reading, or labeling emails). This behavior directly aligns with the trends observed in Fig. 2b, offering complementary evidence that GVPO encourages agents to act more carefully and reliably by consulting external resources prior to execution.

## 5 RELATED WORK

**RLVR.** Since DeepSeek-R1 (Guo et al., 2025), research on the RLVR paradigm has accelerated significantly (Wen et al., 2025; Xie et al., 2025). This line of work covers diverse dimensions such as training data curation (Wang et al., 2024), objective formulation (Liu et al., 2025), hyperparameter

optimization (Yu et al., 2025), base model selection (Hu et al., 2025), and empirical insights (Yue et al., 2025). Verified rewards in prior work are typically derived from deterministic outcome checks (e.g., exact match in math (Cobbe et al., 2021) , unit tests in coding (Austin et al., 2021)), rule-based verification with tools Li et al. (2025); Qian et al. (2025), LLM-based verifiers (Wen et al., 2025; Chen et al., 2024), logic-based verifiers (Wang et al., 2025) or domain-specific reward models (Su et al., 2025). The related work GiGPO (Feng et al., 2025) employs an additive shaping function to integrate step-relative advantages but does not leverage intermediate process feedback. In contrast, we introduce a more general advantage shaping framework that unifies both additive and multiplicative formulations, and validate its effectiveness on the more challenging AppWorld benchmark.

**RL for LLM agents.** A complementary line of research investigates *tool-use learning* (Yao et al., 2023), where agents are trained to interact with external environments through APIs (Qin et al., 2024), code execution (Li et al., 2025), or multi-turn reasoning (Wei et al., 2025; Xi et al., 2025; Da et al., 2025; Mai et al., 2025). The applications span text-based games (Narasimhan et al., 2015; Yao et al., 2020; Carta et al., 2023), web navigation and shopping (Yao et al., 2022), mobile device interaction (Bai et al., 2024), and embodied tasks (Zhai et al., 2024), yet our work focuses on AppWorld (Trivedi et al., 2024), a significantly more challenging benchmark that requires long-horizon, multi-app interactions. The most related effort is LOOP (Chen et al., 2025b), but unlike LOOP, GVPO incorporates signals from intermediate execute feedback, enabling more accurate credit assignment and improved robustness.

## 6 CONCLUSION

We propose Group Verification-based Policy Optimization (GVPO), a reinforcement learning algorithm that unifies outcome-verifiable and process-verifiable signals through an advantage shaping framework. By leveraging intermediate execution feedback alongside final task outcomes, GVPO achieves more accurate credit assignment, resulting in greater training stability, accelerated convergence, and improved generalization in complex interactive environments. Empirical results on the challenging AppWorld benchmark demonstrate that GVPO not only surpasses strong RL baselines but also closes the gap with much larger proprietary systems, highlighting its potential as a scalable approach for training stateful, multi-turn LLM interactive agents.

**Limitations.** GVPO currently relies on deterministic environments with well-defined process signals, and its effectiveness on tasks with ambiguous or noisy feedback remains underexplored. Extending GVPO to broader domains and integrating it with richer supervision sources are important directions for future work.

## ACKNOWLEDGMENTS

The authors wish to thank the reviewers for their helpful comments and suggestions. This work was supported by National Key R&D Program of China (No. 2025ZD1801501), NSFC grant (No. 62136002 and 62477014), Ministry of Education Research Joint Fund Project (8091B042239), Shanghai Knowledge Service Platform Project (No. ZF1213).

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

# A  APPENDIX

## A.1  HYPERPARAMETERS AND TRAINING SETUP

| Setting | Value |
|---|---|
| *Base setup* | |
| Base model | Qwen2.5-32B-Instruct |
| Framework | veRL + FSDP2 + vLLM (batched inference) |
| Training tasks | AppWorld difficulty-1/2 (72 samples, 24 scenarios) |
| Rollouts per sample | 8 |
| Max turns (train / eval) | 40 / 50 |
| Temperature (train / eval) | 1.0 / 0.0 |
| *GVPO hyperparameters* | |
| Group size $G$ | 8 |
| Clipping range ($\epsilon_{low}, \epsilon_{high}$) | (0.2, 0.28) |
| Learning rate | $1 \times 10^{-5}$(constant) |
| Batch size | $4 \times 8$ |
| Max sequence length per turn | 512 tokens |
| Entropy coefficient | 0.0 |
| Advantage shaping penalty $b$ | 0.2 |
| Optimizer | AdamW |
| Training steps | 144 |
| Mirco batchsize | 8 |

Table 5: Hyperparameters and training setup for GVPO experiments in AppWorld.

**Training setup.**  We conduct all experiments on a single machine equipped with 8 NVIDIA H100 GPUs. Our training framework is based on `verl` with `FSDP2` for efficient distributed training. For data generation, we employ `vLLM` to perform rollouts. After collecting trajectories, we recompute the per-token log-probabilities under the generating policy, rather than using the values directly reported by `vLLM`.

**Agent setup.**  All agents are prompted in a ReAct-style format, which includes one in-context example of a successful task execution. The agent receives as input the results of code execution (e.g., API call outputs or exception traces), together with the original task instruction. At each turn, the agent is allowed to generate up to 512 tokens in total. This limit covers both reasoning tokens (chain-of-thought) and code. If an API response exceeds 3K tokens, it is truncated, and the agent is provided with a short note indicating that truncation has occurred.

**AppWorld setup.**  During training, we launch 32 independent AppWorld backend services in advance. The training framework communicates with these AppWorld backends via a Redis message queue, which serves as the central communication hub. Each backend service is bound to a unique port, and port numbers are used to associate each AppWorld instance with its corresponding trajectory. This design ensures that multiple trajectories can be executed in parallel while avoiding interference across different AppWorld backends.

**Training Hyperparameters.**  We use a constant learning rate of $1 \times 10^{-5}$ and clip the gradient norm to 1 in all experiments. Each training sample produces 8 rollouts with temperature 1.0. To accelerate training, we apply early stopping to rollout collection: rollout generation is terminated once at least 6 rollouts have been collected for each task and 90% of the total rollouts have been collected. Concretely, we consider two stopping conditions. First, within each group of 8 rollouts for a given task, rollout collection ends once 6 rollouts have finished. Second, across the 32 sampled tasks, rollout collection terminates once 30 tasks have completed. Early stopping is only applied after the model has generated at least 30 steps of rollout, ensuring sufficient exploration before termination. We allow up to 40 interactions between the agent and the environment during training and up to 50 for evaluation. Any episode that does not complete within this interaction budget is considered a failure. If the sequence reaches the model's context window limit, the rollout is terminated (Tab. 5).

| | Para. | Std. | Aggr. | Out. | Proc. | Dev | |
|---|---|---|---|---|---|---|---|
| | | | | | | TGC | SGC |
| *Fine-tuning with RL* | | | | | | | |
| GRPO$_{w/\,kl}$ | 32B | ✓ | *smtm* | ✓ | ✗ | 75.4 | 63.2 |
| GRPO | 32B | ✓ | *smtm* | ✓ | ✗ | 77.2 | 57.9 |
| GSPO | 32B | ✓ | *smtm* | ✓ | ✗ | 63.2 | 42.1 |
| DAPO | 32B | ✓ | *tm* | ✓ | ✗ | 68.4 | 47.4 |
| Dr.GRPO | 32B | ✗ | *smts* | ✓ | ✗ | 77.2 | 68.4 |
| RLOO$_{w/\,kl}$ | 32B | ✗ | *smtm* | ✓ | ✗ | 68.4 | 57.9 |
| GVPO | 32B | ✗ | *smtm* | ✓ | ✓ | **84.2** | **73.7** |

Table 6: Comparison of RL fine-tuning methods on the AppWorld Dev set. **Para.**: model parameter scale. **Std.**: whether std normalization is applied in advantage computation. **Aggr.**: loss aggregation scheme, where *smtm* = sequence-mean-token-mean, *smts* = sequence-mean-token-sum, and *tm* = token-mean. **Out.**: use of outcome-verifiable signals. **Proc.**: use of process-verifiable signals. GRPO$_{w/\,kl}$ denotes GRPO trained with KL regularization. The best results are shown in bold, and the second-best results are underlined.

## A.2 BASELINES

This section provides a detailed overview of the reinforcement learning (RL) algorithms evaluated in our experiments. Each method represents a different strategy for variance reduction, credit assignment, or stability enhancement in policy gradient optimization.

1. **RLOO (Reinforce Leave-One-Out).** Builds upon the REINFORCE estimator by introducing a leave-one-out baseline within each rollout group. This design reduces variance in advantage estimation compared to vanilla REINFORCE, leading to more stable updates without requiring additional learned value functions.
2. **GRPO (Group Relative Policy Optimization).** Computes relative advantages by normalizing rewards within a rollout group, thereby stabilizing training against reward scale fluctuations. GRPO has become a standard RLVR approach for LLM fine-tuning. Variants may additionally incorporate KL regularization with respect to the base model to control divergence.
3. **GSPO (Group Sequence Policy Optimization).** Moves from token-level to sequence-level optimization by defining the importance ratio at the trajectory level. It applies sequence-level clipping, which simplifies optimization and reduces variance. This approach has demonstrated strong performance and efficiency, particularly in recent Qwen3 models.
4. **Dr.GRPO (GRPO Done Right).** Addresses biases in GRPO by (i) removing the normalization with group standard deviation during advantage computation, and (ii) modifying the loss aggregation scheme to *smts*. These changes improve token efficiency and reduce optimization bias, while preserving the reasoning capability of the model.
5. **DAPO (Decoupled Clip and Dynamic sAmpling Policy Optimization).** Combines *clip-higher* for exploration and a *token-level loss* for fine-grained credit assignment; we omit dynamic sampling and overlong reward shaping.
6. **LOOP (Leave-One-Out PPO).** Extends RLOO by adopting a PPO-style optimization procedure, applying multiple epochs of clipped updates per batch. This combination improves exploration and enhances policy robustness in long-horizon interactive tasks. As the original implementation was not open-sourced, we rely on the reported results from the publication, which are available only for the Test set.

## A.3 VALIDATION RESULT

Tab. 6 presents the updated validation results on the Dev split for all RL fine-tuning methods. Our method, GVPO, achieves the best overall performance, reaching **84.2** TGC and **73.7** SGC, outperforming all baselines by a clear margin. The improvements are substantial: compared with the strongest baseline Dr.GRPO (77.2 TGC / 68.4 SGC), GVPO yields gains of +7.0 TGC and +5.3 SGC. This demonstrates that incorporating process-level verifiable signals provides consistent benefits on both task-level completion and scenario-level consistency.

Among baselines, Dr.GRPO delivers the strongest performance, tying for the highest TGC (77.2) and achieving a relatively high SGC (68.4), indicating that its modified loss aggregation scheme and the removal of standard deviation normalization in advantage computation contribute to more stable and effective optimization. Standard GRPO achieves competitive TGC (77.2) but exhibits noticeably lower SGC (57.9), suggesting that outcome-only supervision is insufficient to ensure scenario-level coherence. Adding KL regularization to GRPO slightly reduces TGC (75.4) but improves SGC (63.2), reflecting a trade-off between exploration and stability. Similarly, GSPO and DAPO achieve comparatively lower results. GSPO replaces token-level importance sampling with sequence-level importance sampling, and DAPO adopts token-mean loss aggregation with a clip-high strategy; these design choices lead to more moderate improvements on this setting.

Overall, the Dev results consistently show that explicitly leveraging process-verifiable feedback leads to stronger and more balanced improvements across both metrics. The clear margin on the development set further confirms that GVPO provides a more informative and stable optimization signal during training, which aligns with the trends observed on the Test split.

### A.4 GENERALIZATION BEYOND APPWORLD

To further examine the generalization ability of GVPO beyond the AppWorld environment, we conduct additional experiments on two widely used interactive reinforcement learning benchmarks: ALFWorld and WebShop. These environments differ substantially from AppWorld in task structure, action space, and intermediate feedback signals, providing a rigorous test of cross-domain robustness.

**Experimental Setup**   For both ALFWorld (Shridhar et al., 2020) and WebShop (Yao et al., 2022), we adopt a unified experimental setting. We train Qwen2.5-7B-Instruct using GVPO and GRPO for 100 training steps under identical hyperparameters. For each method, the best-performing checkpoint is selected based on validation performance and evaluated on the corresponding test set. During interaction, if the environment returns "nothing happens" or the model produces an invalid or improperly formatted action, the step is treated as a failure.

**Results on ALFWorld**   Tab. 7 reports the performance on ALFWorld across different task categories. GVPO consistently outperforms GRPO by a large margin in overall success rate as well as in most individual task types. Notably, GVPO achieves strong gains on challenging categories such as Clean, Heat, and Pick2. Moreover, GVPO substantially surpasses strong proprietary systems, including Gemini-2.5-Pro and GPT-4o, in overall performance.

These results indicate that GVPO effectively leverages step-level feedback to improve long-horizon decision making in embodied interaction tasks.

|  | Pick | Look | Clean | Heat | Cool | Pick2 | All |
|---|---|---|---|---|---|---|---|
| GPT-4o | 75.3 | 60.8 | 31.2 | 56.7 | 21.6 | 49.8 | 48 |
| Gemini-2.5-pro | 92.8 | 63.3 | 62.1 | 69 | 26.6 | 58.7 | 60.3 |
| GRPO | 85.3 | 86.7 | 80 | 15.4 | 65 | 56.2 | 70.3 |
| GVPO | 89.2 | 57.1 | 95.8 | 80 | 85.7 | 82.4 | 84.4 |

Table 7: Experiment result on ALFWorld

**Results on WebShop**   We further evaluate GVPO on the WebShop benchmark, which involves web-based shopping interactions with sparse and noisy feedback. As shown in Tab. 8, GVPO again outperforms GRPO and leading proprietary models. In particular, GVPO achieves a higher success rate than both GPT-4o and Gemini-2.5-Pro, demonstrating consistent advantages in a markedly different interaction setting.

**Analysis**   The above results suggest that the effectiveness of GVPO is not limited to coding-oriented or synthetic environments. Many real-world interactive tasks naturally produce step-level failure signals, such as invalid navigation actions, failed API calls, or inconsistent environment

|  | Score | Success rate |
|---|---|---|
| GPT-4o | 31.8 | 23.7 |
| Gemini-2.5-pro | 42.5 | 35.9 |
| GRPO | 79.7 | 70.3 |
| GVPO | 90.2 | 80.5 |

Table 8: Experiment result on Webshop

states. GVPO explicitly exploits these signals through step-level advantage shaping, enabling more accurate credit assignment and safer policy updates.

Overall, these additional experiments confirm that GVPO generalizes robustly across agents, datasets, and domains, and provides consistent performance gains beyond the original AppWorld setting.

## A.5 EXAMPLE OF APPWORLD

Fig. 4 illustrates a representative trajectory from the AppWorld Spotify task, showcasing how the agent interacts with APIs in a multi-step reasoning process. The agent first queries the supervisor's profile and stored account passwords, then uses this information to successfully authenticate with the Spotify API and obtain an access token. With the valid token, it proceeds to call the recommendation API to retrieve music suggestions. This example highlights how the agent effectively integrates intermediate signals, such as execution success, to guide subsequent actions, ensuring correctness and reliability in a complex, stateful environment.

## A.6 LLM USAGE

In accordance with the ICLR policy on the use of large language models (LLMs), we report that LLMs were used exclusively for language polishing and grammar refinement in this paper. LLMs were not used for designing experiments, conducting analyses, or generating research content. All research ideas, implementations, and results are the work of the authors.

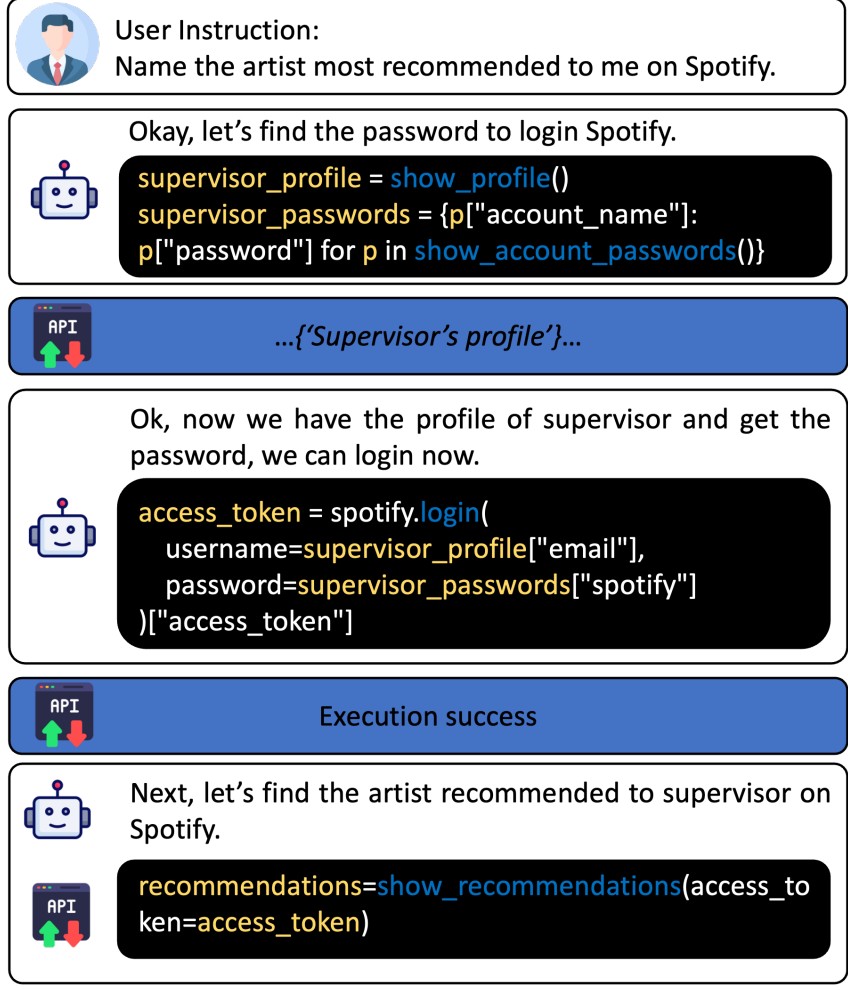

Figure 4: Example trajectory from AppWorld's Spotify task.

