# OpenReview forum: "Group Verification-based Policy Optimization for Interactive Coding Agents"
_ICLR.cc/2026/Conference — ICLR 2026 Poster_

### Official Review · Reviewer_34BC · 2025-10-30

**Soundness:** 3
**Presentation:** 3
**Contribution:** 2
**Rating:** 4
**Confidence:** 3

**Summary:**

The paper proposes GVPO, a reinforcement learning method for interactive code agents. The approach integrates outcome-verifiable signals and process-verifiable signals (e.g., syntax errors, runtime exceptions, partial test passes) into a shaped advantage function, and further stabilizes training through asymmetric clipping and sequence-mean–token-mean aggregation, which improve stability and length fairness.
In the AppWorld environment, GVPO achieves consistent improvements over baselines such as GRPO, Dr.GRPO, DAPO, and LOOP, and provides several ablation studies.

**Strengths:**

1. The stability design is well thought out: asymmetric clipping and two-level aggregation reduce gradient explosion and sequence-length bias.

2. Experiments are relatively comprehensive within a single environment, including main tables and multiple ablations with consistent trends, and the setup appears reproducible.

**Weaknesses:**

- I am not an expert in this subfield, but from my perspective the conceptual novelty is limited: combining intermediate and final feedback for reward shaping is rather straightforward within the RLVR framework (Code Agent).

- The theoretical analysis is insufficien, the paper lacks quantitative discussion of bias/variance changes and convergence properties introduced by advantage shaping.

- Empirically, GVPO performs comparably to LOOP on the normal splits and shows moderate advantage on the challenge split.

- The related-work section does not provide a sufficiently comprehensive overview of prior works, making the claimed contributions difficult to clearly position.

**Questions:**

- How is the shaping coefficient b chosen, and how sensitive is performance to it?


- How sensitive are the results to the chosen upper and lower bounds in asymmetric clipping?


- The authors note that the KL term is removed for GRPO (only comparing with GRPO-w/kl), but the effect of GVPO + KL has not been evaluated.

---

> ### Author Response · Authors · 2025-11-27
>
> We sincerely thank the reviewer for the thoughtful and constructive feedback. Your comments have significantly helped us refine the paper.
> Our detailed rebuttal is provided below.
> ### W1: I am not an expert in this subfield, but from my perspective the conceptual novelty is limited: combining intermediate and final feedback for reward shaping is rather straightforward within the RLVR framework (Code Agent).
> A: We understand the reviewer’s perspective: combining intermediate process feedback with final rewards may appear conceptually straightforward within the RLVR setting. We clarify the distinction between simple reward shaping and GVPO:
> - **Baseline comparison**: Subtracting a fixed penalty from the final reward for execution errors behaves like GRPO, lacking step-level credit assignment.
> - **GVPO advantage**: Modifies advantages at the step level using execution traces, enabling precise credit assignment, reducing failure-prone actions, and improving execution reliability.
> - **Empirical evidence**: GVPO outperforms both GRPO and simple reward shaping, showing that combining intermediate and final feedback at the reward level alone does not achieve these gains.
> ||TGC-N|SGC-N|TGC-C|SGC-C|
> |---|:---:|:---:|:---:|:---:|
> |GVPO|71.4|53.6|49.3|29.5|
> |GRPO|54.8|30.4|35.5|15.8|
> |GRPO + reward shaping (penalty=0.2)|60.1|37.5|39.9|23.0|
>
> This highlights GVPO’s contribution: converting execution traces into process-verifiable, step-level signals that directly shape policy gradients.
>
> ### W2: The theoretical analysis is insufficien, the paper lacks quantitative discussion of bias/variance changes and convergence properties introduced by advantage shaping.
> A: Thank you for the comment. While our focus is empirical, we clarify key theoretical aspects of advantage shaping:
> $A_{i,t} = f(\hat{A_{i,t}}, B_{i,t})$
> - **Bias**: Controlled and local, since shaping modifies step-level advantages, uses a constant coefficient b=0.2, and normalization prevents scaling with trajectory length.
> - **Variance reduction**: Dense step-level signals replace sparse terminal rewards, improving credit assignment.
> - **Convergence intuition**: Under a fixed environment and bounded b, shaping preserves the optimal policy up to a potential shift, analogous to potential-based shaping.
>
> We will expand this theoretical analysis in the appendix to make these effects explicit.
> ### W3: Empirically, GVPO performs comparably to LOOP on the normal splits and shows moderate advantage on the challenge split.
> A: Thank you for the observation. The challenge split (417 tasks) is the primary AppWorld benchmark and is substantially harder than the normal split (168 tasks), with longer interactions, more tools, and cross-App scenarios. Step-level credit assignment has greater impact here, so GVPO’s gains better reflect its benefits in complex tasks.
>
> ### W4: The related-work section does not provide a sufficiently comprehensive overview of prior works, making the claimed contributions difficult to clearly position.
> A: Thank you for the comment. We will reorganize and expand the related-work section to better position GVPO:
> - Distinguish outcome-based verification, LLM/learned reward models, and process-level execution feedback, highlighting GVPO’s use of deterministic step-level signals.
> - Contrast reward shaping vs. advantage shaping, showing GVPO leverages intermediate execution traces unlike prior methods.
> - Discuss tool-use and multi-step agent learning (e.g., AppWorld, LOOP), emphasizing GVPO’s step-level credit assignment for long-horizon tasks.
>
> This clarifies that GVPO’s main contribution is a general advantage-shaping mechanism integrating process-verifiable signals at the step level.

---

> ### Author Response · Authors · 2025-11-27
>
> ### Q1: How is the shaping coefficient b chosen, and how sensitive is performance to it?
> A: Thank you for the question. As detailed in the Global Response (Shaping coefficient b section), we swept b over 0.1, 0.2, and 0.4, showing GVPO’s performance is stable across this range. Future work could explore adaptive or learned shaping.
>
> ### Q2: How sensitive are the results to the chosen upper and lower bounds in asymmetric clipping?
> A: Thank you for the question. We conducted a sensitivity study on asymmetric clipping bounds:
> |upper bound|TGC-N|SGC-N|TGC-C|SGC-C|
> |---|:---:|:---:|:---:|:---:|
> |0.2|59.5|37.5|40.7|20.1|
> |0.24|64.3|37.5|38.9|20.9|
> |0.28|71.4|53.6|49.3|29.5|
> |0.32|64.9|39.3|44.5|24.5|
>
> - **Setup**: Upper bound varied (0.20, 0.24, 0.28, 0.32) with lower bound fixed.
> - **Findings**:
>   - Too tight (0.20): overly constrained updates, weaker performance.
>   - Moderate (0.24–0.28): improved learning, with 0.28 performing best (used in main experiments).
>   - Higher (0.32): still competitive, but slightly less stable.
>
> This analysis is included in the appendix to clarify the effect of asymmetric clipping.
>
> ### Q3: The authors note that the KL term is removed for GRPO (only comparing with GRPO-w/kl), but the effect of GVPO + KL has not been evaluated.
> A: Thank you for the question. We evaluated adding KL regularization to both GRPO and GVPO:
> ||TGC-N|SGC-N|TGC-C|SGC-C|
> |---|---|---|---|---|
> |GRPO|54.8|30.4|35.5|15.8|
> |GRPO+KL|61.3|42.9|38.8|19.4|
> |GVPO|71.4|53.6|49.3|29.5|
> |GVPO+KL|47|26.8|27.2|10.7|
>
> - **GRPO**: KL improves performance by stabilizing updates and reducing drift, especially on simpler tasks.
> - **GVPO**: KL degrades performance because step-level process feedback pushes the policy away from failure-prone behaviors; KL pulls it back toward a suboptimal reference, suppressing corrective updates.
>
> This shows that KL and process-level shaping can conflict in tool-based environments. We will include these results and discuss alternative trust-region approaches (e.g., adaptive or token-level KL) in the revision.

---

### Official Review · Reviewer_JpF7 · 2025-10-31

**Soundness:** 2
**Presentation:** 2
**Contribution:** 3
**Rating:** 6
**Confidence:** 3

**Summary:**

This paper proposes Group Verification-based Policy Optimization (GVPO), a novel reinforcement learning algorithm that integrates outcome-verifiable rewards with process-verifiable feedback. By leveraging both long-term task alignment signals and intermediate corrective signals, GVPO improves credit assignment, optimization stability, and generalization in complex interactive environments. Experimental results demonstrate that a 32B-parameter agent trained with GVPO outperforms strong baselines, including OpenAI's o1 agent, particularly on challenging benchmarks like Test-C in the AppWorld environment.

**Strengths:**

1. The proposed GVPO framework is novel and well-motivated, with a clean and intuitive formulation that combines outcome-verifiable and process-verifiable feedback.
2. By incorporating step-level successes and failures through process-verifiable signals, GVPO effectively constrains optimization, leading to more reasonable and guided learning. The experimental results convincingly support this claim.
3. The paper presents a comprehensive experimental setup, including appropriate ablations and a sufficiently large model size (32B), which adds credibility to the findings.

**Weaknesses:**

1. The paper does not sufficiently discuss the computational cost of GVPO, particularly in terms of scalability to larger models or environments.
2. The experiments are limited to a 32B-parameter model, leaving open the question of how GVPO would perform with larger-scale models, especially given the trend toward scaling in reinforcement learning research.

**Questions:**

1. How does GVPO scale computationally when applied to larger models or more complex environments? Are there practical limitations or bottlenecks?
2. Could GVPO benefit from additional architectural modifications or optimizations specific to larger-scale models (e.g., 64B or beyond)?
3. Would GVPO’s advantage shaping framework generalize to other interactive environments beyond AppWorld? Any insights into potential domain-specific challenges?

---

> ### Author Response · Authors · 2025-11-27
>
> We sincerely thank the reviewer for the thoughtful and constructive feedback. Your comments have significantly helped us refine the paper.
> Our detailed rebuttal is provided below.
> ### W1&W2: The paper does not sufficiently discuss the computational cost of GVPO, particularly in terms of scalability to larger models or environments. The experiments are limited to a 32B-parameter model, leaving open the question of how GVPO would perform with larger-scale models, especially given the trend toward scaling in reinforcement learning research.
> A: Thank you for the comment. GVPO was trained on 8 H100 GPUs with models up to 32B parameters; we have not scaled beyond this due to resource limits.
> GVPO adds only lightweight overhead: execution-trace verification is integrated into environment interaction, and advantage shaping is applied post hoc without extra forward passes or external evaluators. Overall compute primarily scales with model size, with GVPO’s shaping adding minimal overhead.
> We will clarify this and discuss larger-scale applicability as future work.
>
> ### Q1&Q2: How does GVPO scale computationally when applied to larger models or more complex environments? Are there practical limitations or bottlenecks? Could GVPO benefit from additional architectural modifications or optimizations specific to larger-scale models (e.g., 64B or beyond)?
> A: Thank you for the questions. GVPO adds minimal overhead to standard GRPO-style training: execution-trace verification occurs during environment interaction, and advantage shaping is applied post hoc without extra forward passes or external evaluators. Thus, overall compute primarily scales with model size.
> Practical limitations come from GPU memory and runtime for very large models (e.g., 64B+) or complex environments with long trajectories. While we have not tested these scales, GVPO is compatible with standard large-model optimizations (gradient checkpointing, mixed precision, pipeline parallelism), and architecture-specific strategies (e.g., model-parallelism, dynamic batch sizing) could further improve efficiency. These points will be noted in the revision.
>
> ### Q3: Would GVPO’s advantage shaping framework generalize to other interactive environments beyond AppWorld? Any insights into potential domain-specific challenges?
> A: Thank you for the question. As detailed in the global response (Generalization section), GVPO was evaluated on WebShop (web) and ALFWorld (embodied) with 7B models, showing consistent performance gains beyond AppWorld.
> Potential domain-specific challenges include environments where step-level feedback is sparse, noisy, or non-verifiable, such as open-ended reasoning tasks or partially observable settings. In such cases, auxiliary mechanisms like LLM-based judgment or learned validators may be required to maintain effective shaping.

---

### Official Review · Reviewer_N8jC · 2025-11-01

**Soundness:** 3
**Presentation:** 3
**Contribution:** 2
**Rating:** 4
**Confidence:** 4

**Summary:**

This paper introduces Group Verification-based Policy Optimization (GVPO), a novel RL from Verifiable Rewards (RLVR) algorithm designed to enhance the training of interactive coding agents. The core insight is that existing RLVR algorithm rely solely on sparse, outcome-verifiable rewards (e.g., final unit test results), leading to inaccurate credit assignment across an agent's reasoning trajectory. GVPO addresses this by integrating dense, process-verifiable feedback according to the advantage estimation directly into the optimization objective. Experiments shows that GVPO outperforms existing RLVR algorithms on AppWorld, and Qwen2.5-32B-Instruct trained with RLVR outperforms than zero-shot commercial LLMs.

**Strengths:**

The paper is easy to follow and well-organized.

The paper is well-motivated. Constructing step-level signals are of great importance for RLVR algorithms, especially for long-term hard tasks.

Some empirical results are compelling. Achieving a new SOTA on a complex benchmark and outperforming a model like OpenAI's o1 by a significant margin (12.6% on Test-C) with a 32B model is a strong testament to the method's efficacy.

**Weaknesses:**

One of the main claimed contribution, process-verifiable feedback derived from intermediate execution traces (e.g., syntax errors, runtime exceptions), are not implementated in the algorithm.

The proposed shaping function for $B_{i,t}$ and the resulting $A_{i,t}$ is complex and contains several case-based rules. Designing such function for each secnario is diffcult and may cause reward hacking [1].

The experiments are limited to one agent type(CodeAct), one scenario (AppWorld), and one model (Qwen2.5-32B-Instruct), which can not prove the generalization of GVPO. No evidence is given that GVPO helps in other widely-used agent domains (e.g., WebArena, SWEBench).

All experimental results are point estimates from a single run. It is unclear whether the +3.6 % TGC gain over LOOP is statistically significant.

Reference:
[1] Guo D, Yang D, Zhang H, et al. Deepseek-r1: Incentivizing reasoning capability in llms via reinforcement learning[J].Nature, 2025.

**Questions:**

- Why can GVPO prevent the entropy decay in Figure 2?

- Considering that the number of training samples is only 35, I wonder how many gradient steps the training has?

- The proposed shaping function for $B_{i,t}$ and the resulting $A_{i,t}$ is complex and contains several case-based rules. Designing such function for each secnario is diffcult and may cause reward hacking.

---

> ### Author Response · Authors · 2025-11-27
>
> We sincerely thank the reviewer for the thoughtful and constructive feedback. Your comments have significantly helped us refine the paper.
> Our detailed rebuttal is provided below.
> ### W1: One of the main claimed contribution, process-verifiable feedback derived from intermediate execution traces (e.g., syntax errors, runtime exceptions), are not implementated in the algorithm.
> A: Thank you for the comment. The perceived gap may arise because, in our current implementation, syntax errors, runtime exceptions, and other invalid actions are treated uniformly as failure signals rather than being differentiated into finer-grained categories. This design keeps the verifier simple and consistent but does not change the fact that execution-trace-based feedback is fully integrated.
>
> In practice, these errors are detected during execution and converted into step-level penalties via the environment’s verifier signals, which are then incorporated into the advantage computation for policy updates directly corresponding to the process-verifiable feedback described as a contribution.
>
> To clarify this, we will revise the paper to (i) explicitly describe how execution-trace signals are collected and unified into failure penalties, (ii) provide pseudocode (in the appendix) showing where these signals are integrated in the policy update loop, and (iii) include illustrative examples mapping different error types to reward signals.
>
> ### W2: The proposed shaping function $B_{i,t}$ for and the resulting $A_{i,t}$ is complex and contains several case-based rules. Designing such function for each secnario is diffcult and may cause reward hacking
> A: Thank you for the comment. While the formulas for $B_{i,t}$ and $A_{i,t}$ may appear complex, the underlying principle is simple: we perform advantage shaping based on the original advantage and step-level execution outcomes, and the same mechanism applies broadly wherever environment feedback is available. It is not hand-crafted for each specific task.
> In our implementation, a unified shaping rule is applied rather than case-specific heuristics:
> - If the original advantage is 0, we apply an additive penalty of -b;
> - If the original advantage is positive, we set it to 0;
> - If the original advantage is negative, we amplify it by multiplying by (1+b).
>
> This mechanism applies broadly across environments and relies on objective execution feedback (invalid calls or runtime errors) rather than heuristics of agent behavior, limiting reward-hacking opportunities.
> We will clarify this rationale and improve the presentation of the shaping rule in the revised paper.
> ### W3: The experiments are limited to one agent type(CodeAct), one scenario (AppWorld), and one model (Qwen2.5-32B-Instruct), which can not prove the generalization of GVPO. No evidence is given that GVPO helps in other widely-used agent domains (e.g., WebArena, SWEBench).
> A: Thank you for the comment. We address generalization in the global response (Generalization section), where we provide new experiments on WebShop and ALFWorld, which use the React agent rather than CodeAct and demonstrate strong gains beyond the original AppWorld setting. These experiments were conducted with Qwen2.5-7B-Instruct, showing that GVPO’s benefits extend across different agents, environments, and base model sizes.
>
> ### W4: All experimental results are point estimates from a single run. It is unclear whether the +3.6 % TGC gain over LOOP is statistically significant.
> A: Thank you for the comment. We repeated GVPO and LOOP on AppWorld with three independent seeds and report the mean and standard deviation of the best checkpoint on Test C. Across all runs, GVPO consistently outperforms LOOP, showing that the +3.6% TGC gain is robust and not due to single-run variance. These results are included in the revised paper.
> ||Test C, TGC (mean ± std across 3 seeds)|
> |---|:---:|
> |LOOP|45.7 ± 1.3|
> |GVPO|49.3 ± 2.3|

---

> ### Author Response · Authors · 2025-11-27
>
> ### Q1: Why can GVPO prevent the entropy decay in Figure 2?
> A: Thank you for the question. We computed the proportion of clip-high events over the first 40 micro batches to understand GVPO’s slower entropy decay:
> ||1-5|6-10|11-15|16-20|21-25|26-30|31-35|36-40|
> |---|---|---|---|---|---|---|---|---|
> |GVPO|0.0230|0.0036|0.0015|0.0011|0.0022|0.0011|0.0017|0.0013|
> |DAPO|0.0294|0.0055|0.0023|0.0120|0.0104|0.0077|0.0026|0.0018|
> |Δ%|+27.8%|+52.8%|+53.3%	|+991%|+372.7%|+600%|+52.9%|+38.5%|
> - **Proportion of clip-high tokens**: DAPO consistently shows a higher fraction of clip-high events than GVPO (e.g., 0.0120 vs. 0.0011 in batches 16–20), with relative differences from ~28% to over 900%.
> - **Effect on entropy**: Recent findings[1] shows clip-high events suppress entropy, while clip-low events tend to increase it.
> - **Interpretation**: Since GVPO encounters far fewer clip-high events, its policy avoids premature exploitation, naturally slowing entropy collapse.
>
> This analysis explains why GVPO maintains higher entropy, and we have added it to the revised manuscript.
>
> Reference:[1] Park J. R., Kim J., Kim G., et al. Clip-Low Increases Entropy and Clip-High Decreases Entropy in Reinforcement Learning of Large Language Models, arXiv preprint arXiv:2509.26114, 2025.
>
> ### Q2: Considering that the number of training samples is only 35, I wonder how many gradient steps the training has?
> A: Thank you for the question. Although AppWorld has only 35 scenarios, training uses repeated rollouts, giving a much larger effective sample size. Following the official split (3 tasks per scenario) yields 72 training tasks (levels 1–2). Each task is rolled out 8 times per update over 8 epochs with early stopping.
> |Quantity|Value|Notes|
> |---|---|---|
> |# Scenarios|35|Official dataset split|
> |# Task instances|105|3 per scenario|
> |# Training tasks used|72|Difficulty 1 & 2|
> |Rollouts per task|8|Per update|
> |Epochs|8|Online RL + early stopping|
>
> The effective number of sampled trajectories is calculated as:
>
> $\text{Effective trajectories} = 72 \text{ tasks} \times 8 \text{ rollouts/task} \times 8 \text{ epochs} \approx 4608$
>
> Thus, the training signal comes from thousands of trajectories generated online under the evolving policy. These details and the table are included in the revised manuscript.
> ### Q3: The proposed shaping function $B_{i,t}$ for and the resulting $A_{i, t}$ is complex and contains several case-based rules. Designing such function for each secnario is diffcult and may cause reward hacking
> A: Thank you for raising this point. This concern is addressed in our response to W2 above, where we clarify that the shaping function applies a unified, step-level rule based on verifiable execution outcomes, limiting opportunities for reward hacking while remaining consistent across scenarios.

---

### Official Review · Reviewer_1bHw · 2025-11-03

**Soundness:** 3
**Presentation:** 3
**Contribution:** 2
**Rating:** 6
**Confidence:** 4

**Summary:**

This paper proposes Group Verification-Based Policy Optimization (GVPO), a new reinforcement learning (RL) algorithm designed to improve the training of large language model (LLM)–based interactive coding agents. GVPO extends prior Group Relative Policy Optimization (GRPO) by introducing an advantage shaping framework that combines two complementary reward types:

Outcome-verifiable rewards -- reflecting final task correctness (e.g., unit-test results);

Process-verifiable signals -- reflecting intermediate feedback such as syntax errors or runtime exceptions.

The method shapes per-step advantages using both signals to achieve finer credit assignment and more stable optimization. Evaluations on AppWorld, a challenging multi-turn code-execution benchmark, show that a 32B-parameter GVPO agent outperforms OpenAI’s o1 by 12.6% on the hardest test split and the strongest 32B RL baseline (LOOP) by 3.6%. Analyses indicate better exploration stability, reduced execution errors, and more cautious agent behavior.

**Strengths:**

1. The integration of process-verifiable signals addresses the sparsity and delay of traditional outcome-only RL signals, providing a principled improvement to credit assignment.
2. GVPO achieves state-of-the-art performance among open 32B-scale agents on AppWorld, significantly surpassing both outcome-only RL methods and prompting-based models.
3. Ablation and behavior analyses (entropy, cautiousness, failure rates) convincingly support the claimed benefits in stability and exploration.
4. Since process-verifiable feedback is rule-based, GVPO avoids additional reward-model training and can generalize to other deterministic environments.

**Weaknesses:**

1. Experiments focus solely on the AppWorld coding environment; it remains unclear how GVPO performs in non-deterministic or partially verifiable domains (e.g., reasoning or natural dialogue tasks).
2. Lack of comparison to LLM-based verifiers: Recent RLVR works using LLM verifiers are not empirically compared, which limits positioning relative to that line of research.
3. While the paper observes “cautious behavior,” it lacks deeper qualitative discussion of how process feedback influences reasoning patterns.
4. The method assumes reliable step-level feedback (syntax/runtime correctness), which may restrict applicability to highly structured environments.

**Questions:**

1. Can this method be applied to other problems beyond coding agent RL training?
2. How sensitive is GVPO to the choice of shaping coefficient b? Could adaptive or learned shaping improve stability across domains?
3. Could the asymmetric clipping mechanism interact with shaping to introduce bias in long trajectories?
4. How well does GVPO scale beyond 32B parameters or to more diverse environments such as web or embodied agents?

---

> ### Author Response · Authors · 2025-11-27
>
> We sincerely thank the reviewer for the thoughtful and constructive feedback. Your comments have significantly helped us refine the paper.
> Our detailed rebuttal is provided below.
> ## W1: Experiments focus solely on the AppWorld coding environment; it remains unclear how GVPO performs in non-deterministic or partially verifiable domains (e.g., reasoning or natural dialogue tasks).
> A: Thank you for the question. As detailed in the global response (Generalization section), we add new experiments on ALFWorld and WebShop, two domains that differ significantly from AppWorld in action space, task structure, and verification complexity. The results show that GVPO generalizes well beyond coding environments.
> For non-deterministic, open-ended domains such as reasoning or dialogue, suitable validators (e.g., learned or LLM-based) are required for step-level verification, which we outline as a promising future direction.
> ## W2: Lack of comparison to LLM-based verifiers: Recent RLVR works using LLM verifiers are not empirically compared, which limits positioning relative to that line of research.
> A: Thank you for the suggestion. We agree that comparisons to LLM-based verifier methods (e.g., RLVR) would add context. Our clarification is as follows:
> - **Resource constraints**: RLVR-style baselines require a large external LLM to serve as the verifier, whose inference cost often exceeds training the policy itself. This made a full empirical comparison infeasible under our submission’s computational budget.
> - **Positioning in related work**: We will revise the related-work section to more clearly contrast GVPO with RLVR approaches, including model-free step-level shaping vs. model-based reward generation, and the associated trade-offs.
> Incorporating learned or LLM-based feedback is a promising hybrid direction, which we plan to explore in future work.
> ## W3: While the paper observes “cautious behavior,” it lacks deeper qualitative discussion of how process feedback influences reasoning patterns.
> A: Thank you for the suggestion. We have added a clearer qualitative analysis showing how process feedback shapes reasoning. In summary:
> 1. **Discourages risky actions**: Step-level penalties catch invalid tool calls and raise the threshold for taking high-risk steps.
> 2. **Encourages information gathering**: As risky actions become less attractive, documentation-query actions gain utility, leading to a “query-before-act” strategy and fewer corrective loops.
> 3. **Improves credit assignment**: Fine-grained rewards pinpoint specific failure modes, enabling more targeted adjustments than sparse scenario-level signals.
>
> Overall, process feedback suppresses unsafe behavior, promotes structured information seeking, and shifts the model toward more deliberate, verify-then-act reasoning.
> ## W4: The method assumes reliable step-level feedback (syntax/runtime correctness), which may restrict applicability to highly structured environments.
> A: GVPO relies on partially verifiable step-level signals (e.g., syntax/runtime correctness), which naturally fit structured domains such as tool-augmented agents, code execution tasks, and API-based workflows.
> For less structured settings, an extension is to incorporate learned or LLM-based evaluators as auxiliary feedback, which we view as a promising direction for broader applicability.
> ## Q1: Can this method be applied to other problems beyond coding agent RL training?
> A: As detailed in the global response (Generalization section), we report additional results on non-coding domains (web shop and embodied agent), showing that GVPO extends beyond coding-agent RL. For less structured settings, suitable validators or learned feedback would be needed, which we leave as future work.
> ## Q2: How sensitive is GVPO to the choice of shaping coefficient b? Could adaptive or learned shaping improve stability across domains?
> A: We address the sensitivity of GVPO to the shaping coefficient b in detail in the global response (**Shaping coefficient b** section), including a sweep across multiple values and discussion of potential adaptive or learned shaping.
> ## Q3: Could the asymmetric clipping mechanism interact with shaping to introduce bias in long trajectories?
> A: In practice, the interaction between asymmetric clipping and step-level shaping introduces limited bias because:
> 1. shaping only applies to local penalties rather than accumulating over full trajectories,
> 2. clipping caps large positive advantages while shaping mainly suppresses negative ones from failure modes.
> Empirically, we observed no reward inflation or trajectory drift. We will clarify this and note that a more formal analysis in long-horizon settings is a valuable direction for future work.

---

> ### Author Response · Authors · 2025-11-27
>
> ### Q4: How well does GVPO scale beyond 32B parameters or to more diverse environments such as web or embodied agents?
> A: Thank you for the question. As detailed in the global response (Generalization section), we report results on ALFWorld and WebShop, showing that GVPO generalizes to both embodied and web-based environments. Scaling beyond 32B was not feasible under our current compute, but studying larger models is an important direction for future work

---

### Author Response · Authors · 2025-11-27
**Global Response**

We thank all reviewers for their constructive feedback. Below we summarize our responses to two common concerns:
1. **Generalization beyond AppWorld**
2. **Sensitivity of the shaping coefficient b**
## Generalization beyond AppWorld
We thank the reviewers for raising questions about GVPO’s generality. Our findings are summarized below.
### New experiments on ALFWorld and WebShop
To directly evaluate generalization, we ran new experiments on two widely used interactive RL benchmarks.
- Shared experimental setup across ALFWorld and WebShop
  - Both benchmarks are **substantially different from AppWorld** in task structure, action modalities, and intermediate feedback signals.
  - We trained **Qwen2.5-7B-Instruct with GVPO and GRPO for 100 steps** under identical settings and evaluated the best checkpoint on each benchmark’s test set.
- ALFWorld
  - GVPO consistently outperforms GRPO by a large margin across all metrics.
  - It also improves most individual task categories (e.g., Clean: 95.8%, Heat: 80.0%, Pick2: 82.4%) and further exceeds strong proprietary systems such as Gemini-2.5-Pro (60.3) and GPT-4o (48.0).
||Pick|Look|Clean|Heat|Cool|Pick2|All|
|---|---|---|---|---|---|---|---|
|GPT-4o|75.3|60.8|31.2|56.7|21.6|49.8|48|
|Gemini-2.5-pro|92.8|63.3|62.1|69|26.6|58.7|60.3|
|GRPO|85.3|86.7|80|15.4|65|56.2|70.3|
|GVPO|89.2|57.1|95.8|80|85.7|82.4|84.4|
- WebShop
  - GVPO again surpasses GRPO and leading proprietary models (GPT-4o: 23.7%, Gemini-2.5-Pro: 35.9%), demonstrating consistent gains across a very different web-shopping interaction setting.
||Score|Success rate|
|---|:---:|:---:|
|GPT-4o|31.8|23.7|
|Gemini-2.5-pro|42.5|35.9|
|GRPO|79.7|70.3|
|GVPO|90.2|80.5|
- Anaysis
  - The results show that **step-level advantage shaping** is not specific to coding environments.
  - Many interactive tasks naturally generate step-level failure signals (e.g., invalid navigation, failed API calls, inconsistent states). GVPO leverages these signals to guide safer and more effective decision making.
  - This explains the consistent improvements across structurally different environments.
- Overall, the new experiments confirm that GVPO **generalizes robustly across agents, datasets, and domains**, providing clear benefits beyond the original AppWorld setting.

## Shaping coefficient b
We thank the reviewers for asking about the choice and sensitivity of the shaping coefficient b. Our findings are summarized below.
### Sensitivity study
We performed a sweep over **b ∈ {0.1, 0.2, 0.4}** on Test-N and Test-C.
|b|TGC-N|SGC-N|TGC-C|SGC-C|
|---|:---:|:---:|:---:|:---:|
|0.1|66.7|39.3|53.1|30.9|
|0.2|71.4|53.6|49.3|29.5|
|0.4|67.9|44.6|48|28.1|
### Key observations
- **b = 0.2** provides the best overall performance and is therefore used for model selection.
- **b = 0.1** weakens step-level corrections and reduces scenario-level performance.
- **b = 0.4** over-penalizes failed steps, introduces mild instability, and does not improve final results.
- Across the tested range, GVPO remains **reasonably stable**, indicating **moderate and manageable sensitivity**.
### Adaptive shaping
We agree that it is promising. One possible extension is to introduce an auxiliary prediction head that estimates b dynamically, allowing the coefficient to adjust based on uncertainty or environmental signals rather than being fixed. We will leave this for future work.

---

### Meta-Review · Area_Chair_RaZE · 2026-01-07

**Summary:**

The paper proposes GVPO, an RLVR method for interactive coding agents that shapes per step advantages using both outcome verifiable rewards and process verifiable execution feedback. On AppWorld, a 32B GVPO agent improves over strong RL baselines and reports gains over o1 on the hardest split. Reviewers generally find the method well motivated and results promising. Main concerns are limited novelty, missing comparisons to LLM verifier based RLVR, generalization, and statistical rigor. The rebuttal adds new generalization results, sensitivity studies, and multi seed runs.

**Reviewer Concerns:**

Addressed by rebuttal:
- Generalization beyond AppWorld: added ALFWorld and WebShop results showing consistent gains over GRPO and strong proprietary baselines.
- Sensitivity to shaping coefficient b and clipping bounds: provided sweeps and a clear best setting.
- Statistical significance: added three seed results on Test C with mean and std.
- Clarified process feedback implementation and provided a simpler description of the shaping rule.
- Added explanation for slower entropy decay with supporting analysis.

Still outstanding:
- Novelty is incremental for some reviewers, framing and related work need tightening.
- No direct empirical comparison to RLVR methods using LLM verifiers, only a cost based justification.
- Limited analysis of failure modes and qualitative behavior changes beyond a high level description.
- Scaling beyond 32B remains untested.
- Some results remain strongest on the challenge split, closer on easier splits.

**Reviewer Scores:**

Two reviewers are above threshold. Two are slightly below but open to acceptance. With the added experiments and multi seed results, borderline reviewers likely move up slightly, but not all the way.

---

### Decision · Program_Chairs · 2026-01-26

Accept (Poster)